# STREAMING VIDEO QUESTION-ANSWERING WITH IN-CONTEXT VIDEO KV-CACHE RETRIEVAL

**Shangzhe Di**[1,2]    **Zhelun Yu**[2]    **Guanghao Zhang**[2]    **Haoyuan Li**[2]    **Tao Zhong**[2]
**Hao Cheng**[2]    **Bolin Li**[2]    **Wanggui He**[2]    **Fangxun Shu**[2]    **Hao Jiang**[2]

[1]Shanghai Jiao Tong University    [2]Alibaba Group
dishangzhe@sjtu.edu.cn

## ABSTRACT

We propose **ReKV**, a novel training-free approach that enables efficient streaming video question-answering (StreamingVQA), by seamlessly integrating with existing Video Large Language Models (Video-LLMs). Traditional VideoQA systems struggle with long videos, as they must process entire videos before responding to queries, and repeat this process for each new question. In contrast, our approach analyzes long videos in a streaming manner, allowing for prompt responses as soon as user queries are received. Building on a common Video-LLM, we first incorporate a sliding-window attention mechanism, ensuring that input frames attend to a limited number of preceding frames, thereby reducing computational overhead. To prevent information loss, we store processed video key-value caches (KV-Caches) in RAM and disk, reloading them into GPU memory as needed. Additionally, we introduce a retrieval method that leverages an external retriever or the parameters within Video-LLMs to retrieve only query-relevant KV-Caches, ensuring both efficiency and accuracy in question answering. ReKV enables the separation of video encoding and question-answering across different processes and GPUs, significantly enhancing the efficiency of StreamingVQA. Through comprehensive experimentation, we validate the efficacy and practicality of our approach, which significantly boosts efficiency and enhances applicability over existing VideoQA models.

## 1 INTRODUCTION

In the literature, video understanding tasks, such as action recognition (Caba Heilbron et al., 2015; Goyal et al., 2017; Kay et al., 2017), visual object tracking (Huang et al., 2019; Muller et al., 2018), and video question-answering (Xu et al., 2017; Jang et al., 2017; Xiao et al., 2021; Li et al., 2024b), have primarily focused on short clips lasting from a few seconds to minutes. However, as vision models increasingly find applications in real-world scenarios like robotics, surveillance, and live broadcasts, the research in the vision community has gradually shifted towards understanding continuous video streams, where long-term contexts and real-time interaction are crucial.

In this paper, we consider the problem of **streaming video question-answering (StreamingVQA)**. As shown in Figure 1(a), it involves continuously processing long video streams and promptly responding to queries about the visual content at any moment. It can be treated as a generalization of the standard offline VideoQA, where the model processes the entire video and all questions simultaneously. By definition, such task of StreamingVQA presents three core challenges: (i) **Efficient Video Encoding:** Unlike traditional offline VideoQA, where models have access to the entire video clip, StreamingVQA demands real-time analysis of continuous streams. Models must efficiently process incoming frames without access to future frames or frequent revisiting of distant past frames. (ii) **Video Context Preservation:** To accurately answer questions posed later in the stream, models must preserve relevant information from earlier frames, making long-term context retention a key challenge. (iii) **Real-Time Response:** The model must provide accurate answers with minimal delay, requiring efficient retrieval of video context and rapid question-answering.

Current Video-LLMs often struggle to encode long video streams due to the large volume of video tokens, forcing most models to process only a sparse subset of frames (Maaz et al., 2024; Zhang

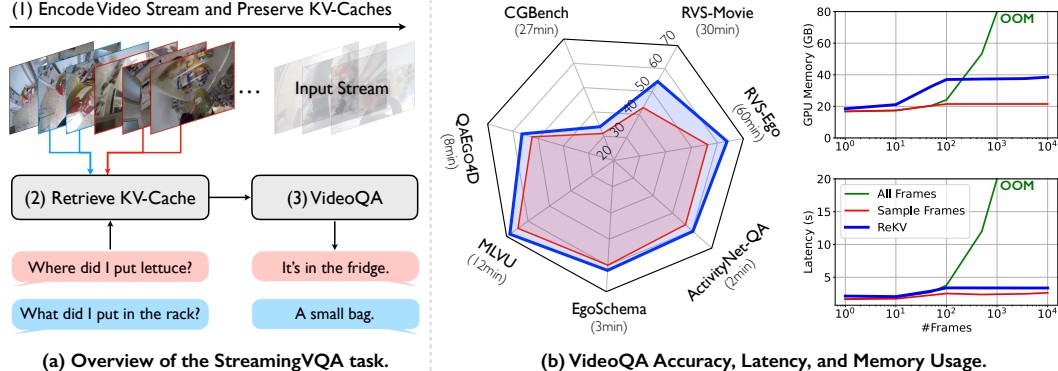

**(a) Overview of the StreamingVQA task.**  **(b) VideoQA Accuracy, Latency, and Memory Usage.**

Figure 1: **Overview of the StreamingVQA task and our proposed ReKV.** (a) StreamingVQA requires a model to continuously process video streams and answer questions about previously viewed content at any moment. (b) We propose ReKV to enhance efficiency and accuracy in StreamingVQA. Tested with `LLaVA-OV-7B` on an H800 (80GB) GPU, ReKV maintains stable latency and GPU memory usage, preventing out-of-memory (OOM) errors as frames increase. It also improves the accuracy on seven long-form VideoQA benchmarks compared to the uniform sampling baseline. Further details are provided in Section 4.

et al., 2024c; Li et al., 2024a). This results in limited video lengths or a significant loss of fine-grained visual information. While techniques like average pooling (Li et al., 2024c) and memory compression (Wu et al., 2022; Wang et al., 2023; He et al., 2024; Zhang et al., 2024a; Qian et al., 2024) reduce token volume, they come at the cost of losing details, particularly in temporal and lower-level visual features that are essential for complex question answering.

To address the challenges, we propose **ReKV** (**Re**trieve In-context Video **KV**-Cache), a framework that seamlessly integrates with existing Video-LLMs (Maaz et al., 2024; Zhang et al., 2024c; Li et al., 2024a) without additional training. Our method employs two strategies for aggregating both short- and long-term temporal information. For **short-term temporal context**, the model adopts causal attention with a sliding-window mechanism (Han et al., 2023), where tokens attend only to a limited set of preceding tokens during encoding. For **recalling long-term information**, we enable dynamic access to any point within the video sequence via retrieval. Specifically, our method retains and reuses past computations (KV-Cache) to avoid redundant processing while enhancing long-term reasoning without sacrificing detail. For extremely long videos, KV-Caches can be offloaded to RAM or disk to prevent memory overflow.

To ensure real-time and accurate responses, we retrieve a fixed number of KV-Caches relevant to the current question. This design strikes a balance between efficiency and accuracy by avoiding the need to process all past frames, while still accessing the most critical information. We experimented with two retrieval methods: one using external CLIP-like models (Radford et al., 2021; Zhai et al., 2023) for semantic matching, and another leveraging internal attention weights for faster, more integrated, and potentially stronger retrieval (Xiao et al., 2024a; Li et al., 2024d).

In summary, ReKV efficiently encodes long video streams, preserves and retrieves in-context KV-Caches to address complex video question-answering. In addition, ReKV separates video encoding from question-answering into distinct processes, further enhancing efficiency. As shown in Figure 1(b), ReKV improves VideoQA accuracy while maintaining stable inference latency and memory usage as frames increase. The remainder of the paper is organized as follows: Section 5 provides an overview of the relevant literature. Section 3 formulates the StreamingVQA task and describes our proposed method in detail. In Section 4, we present ablation studies and comparisons to validate our approach. Consequently, our approach not only enhances accuracy on long VideoQA benchmarks, including MLVU (Zhou et al., 2024a), QAEGO4D$_{MC}$ (Di & Xie, 2024), EgoSchema (Mangalam et al., 2023), and ActivityNet-QA (Yu et al., 2019), as well as StreamingVQA benchmarks (Zhang et al., 2024a), but also reduces inference latency and memory usage.

## 2    STREAMINGVQA: TASK DEFINITION AND DISCUSSION

This paper considers the problem of streaming video question-answering (**StreamingVQA**), where a model continuously processes a video stream and can respond to questions about past visual content at any moment. In this section, we formally define the task and outline the design principles for our proposed solution.

Given a video stream $\mathcal{V}^T := [v_1, v_2, ..., v_T]$ consisting of $T$ frames and a set of $N$ questions $\mathcal{Q} := \{q_1, q_2, \ldots, q_N\}$, StreamingVQA aims to answer a question $q_i$ at any time step $t$ $(1 \leq t \leq T)$, using only the frames seen up to that point, $\mathcal{V}^t := [v_1, v_2, ..., v_t]$.

**Discussion-I: StreamingVQA _vs_. OfflineVQA.** StreamingVQA involves continuously analyzing an incoming video stream and answering questions based on the observed visual content at any moment. In contrast, conventional video question-answering models (Yang et al., 2022; Maaz et al., 2024; Zhang et al., 2024c; Li et al., 2024a) operate in an offline mode, referred to as OfflineVQA. The two paradigms differ in that: 1) StreamingVQA processes a continuous video stream, while OfflineVQA handles a predefined video input, and 2) StreamingVQA allows questions to be asked at any point during the stream, whereas OfflineVQA processes questions only after the entire video has been viewed. Notably, OfflineVQA can be considered a special case of StreamingVQA, where all questions are posed after the video is fully processed.

Conventional approaches typically employ a visual encoder (Radford et al., 2021; Zhai et al., 2023; Fang et al., 2023) and a projection module (Zhang et al., 2024c; Li et al., 2023) to process video frames ($\mathcal{V}^t$). The output is concatenated with the tokenized question to form a sequence $[\mathcal{V}_t, q_i]$ [1], which is then passed to an LLMs to predict an answer. However, this approach is impractical due to the high computational cost associated with processing a large number of frames ($T$).

A common workaround is sparse frame sampling (Maaz et al., 2024; Zhang et al., 2024c; Li et al., 2024a), but this introduces new problems: (i) loss of critical visual information, leading to incomplete or inaccurate responses, and (ii) the need to reprocess frames for different questions, since questions asked at different time points require distinct frame samples. This becomes increasingly inefficient as $T$ and $N$ grow.

Given these challenges, current OfflineVQA methods fall short when applied to StreamingVQA scenarios. Therefore, designing a new approach optimized for StreamingVQA is crucial to handling video streams more efficiently, enabling real-time question answering and unlocking more interactive video analysis applications.

**Discussion-II: Design Principles for Efficient StreamingVQA.** To tackle the aforementioned challenges, we can exploit the causal nature of the LLM decoder to avoid redundant computations and strike a balance between accuracy and speed. During attention calculations, causal masking prevents the model from accessing future tokens, ensuring that video tokens are encoded independently of the questions. This allows us to _decouple video encoding from question-answering_.

For video encoding, we leverage the KV-Cache optimization to accelerate inference. However, as number of frames grows large, handling the massive number of video tokens becomes increasingly inefficient and may exceed the model's capacity (Han et al., 2023; Xiao et al., 2024b). To address this, we adopt a sliding-window attention mechanism (Han et al., 2023), which limits the attention scope to only the most recent frames.

Regarding question-answering, Video KV-Caches are stored and can be reused as context to answer different questions. However, long video sequences produce a substantial amount of KV-Caches, leading to excessive GPU memory consumption, computational overhead, and unnecessary distractions if all are used. To address this, we introduce an efficient retrieval method that selects the most relevant video key-value vectors from the video KV-Caches. These selected vectors then serve as context, enabling efficient and scalable StreamingVQA.

---

[1] We maintain the original notation for simplicity.

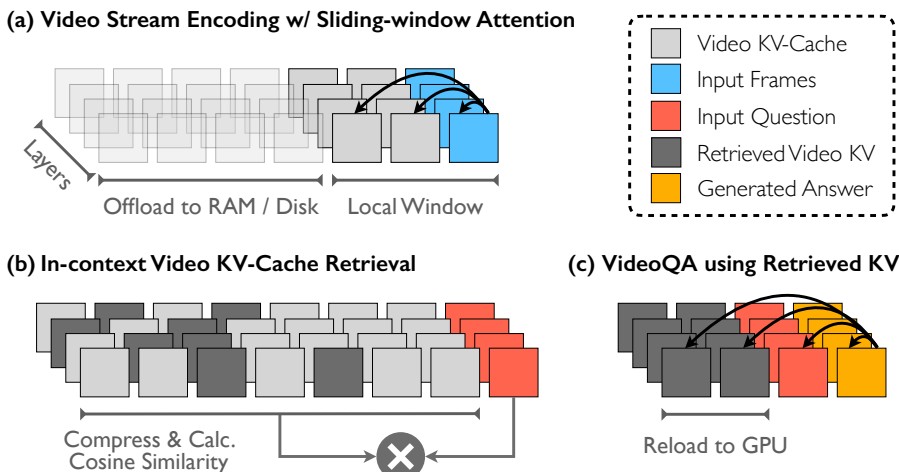

Figure 2: **Overview of ReKV.** We modify the attention mechanism in Decoder-based Video-LLMs: (a) The video stream is encoded with sliding-window attention (Equation 1), with out-of-window Video KV-Caches offloaded to RAM or disk. (b) Upon receiving a question, relevant key-value vectors are retrieved based on cosine similarity, with compressed vectors to accelerate retrieval (Equation 2). (c) The retrieved key-value vectors are reloaded onto the GPU and utilized for autoregressive answer generation (Equation 3).

## 3   ReKV: Retrieve In-context Video KV-Cache

This section introduces **ReKV**, an approach that integrates seamlessly with a Video-LLM to enable efficient StreamingVQA without requiring additional training. Overall, ReKV efficiently encodes the video stream, maintains its KV-Caches, retrieves relevant caches based on the given question, and uses them for accurate question-answering.

We aim to enable Video-LLMs, trained on limited frames, to perform StreamingVQA **without additional training**. As shown in Figure 2, the proposed ReKV has three components: video stream encoding, video KV-Cache retrieval, and question-answering using the retrieved key-value vectors.

**Video Stream Encoding with Sliding-window Attention.** We encode the video stream $\mathcal{V}^T$ incrementally, processing it chunk by chunk. At each step, the inputs include past key-value vectors $\mathbf{P} = \{(\mathbf{k}_j, \mathbf{v}_j)\}_{j=1}^{l_P}$ and the current tokens $\mathbf{X} = \{\mathbf{t}_{i+l_P}\}_{i=1}^{l_X}$, where $l_P$ denotes the lengths of past key-values, and $l_X$ refers to the chunk size. The local key-value vectors within a window $l_L$ can thus be derived as $\mathbf{L} = \mathbf{P}_{[l_P-l_L+1:l_P]}$. The attention calculation is then formulated as:

$$\mathbf{O} = \text{Attn}\left(\mathbf{W_Q X}, [\mathbf{L}_k, \mathbf{W_K X}], [\mathbf{L}_v, \mathbf{W_V X}]\right), \qquad (1)$$

where $\mathbf{W_Q}$, $\mathbf{W_K}$, and $\mathbf{W_V}$ are the attention layer parameters, $\mathbf{L}_k$ and $\mathbf{L}_v$ correspond to the key and value vectors in $\mathbf{L}$. All video KV-Caches are stored for future retrieval. For extremely long videos, we manage memory constraints by offloading KV-Caches to RAM or disk, as in (Xiao et al., 2024a).

**External Video KV-Cache Retrieval.** Here, we utilize an external CLIP-like model (Radford et al., 2021; Zhai et al., 2023) to retrieve question-relevant video KV-Cache, primarily as a baseline to assess whether retrieval can enhance VideoQA performance, as demonstrated in Section 4. Specifically, a CLIP-like model transformers each video frame into a vector $\mathbf{v} = f_v(v) \in \mathrm{R}^D$, where $f_v$ represents the visual encoder, $D$ denotes the vector dimension. Similarly, the question is encoded as $\mathbf{q} = f_t(q) \in \mathrm{R}^D$, where $f_t$ is the text encoder. We then compute the cosine similarity between the embeddings of frame and question:

$$\text{Sim}(\mathbf{v}, \mathbf{q}) = \frac{\mathbf{v} \cdot \mathbf{q}}{\tau \, ||\mathbf{v}|| \, ||\mathbf{q}||} \qquad (2)$$

where $\tau$ is a learnable temperature parameter. This similarity is calculated at the frame level, rather than at the token level. Alternatively, we can group $b$ consecutive frames into blocks by averaging their frame vectors and then compute block-level similarity scores. Finally, the $r$ most relevant video frames or $\lceil r/b \rceil$ video blocks are retrieved. The corresponding video KV-Cache, denoted as $\mathbf{R}$, is subsequently loaded onto the GPU for question-answering.

**Internal Video KV-Cache Retrieval.** Building on recent advancements in handling long sequences with LLMs (Xiao et al., 2024a; Li et al., 2025; Fountas et al., 2025), we further explore using self-attention layers within Video-LLMs for retrieval. Similar to external retrieval, internal retrieval is still performed at the level of video frames or blocks.

During video modeling, the average of the key vectors of a frame is computed as its representative frame vector: $\mathbf{v} = \frac{1}{N_f} \sum_{j=1}^{N_f} \mathbf{k}_j \in \mathrm{R}^{D'}$, where $N_f$ is the number of tokens per frame, and $\mathbf{k}_j$ is the $j$-th key vector. To reduce computational overhead, we do not differentiate between attention heads and instead concatenate them into a single vector, with $D'$ as the resultant dimension. Similarly, the question vector is computed as $\mathbf{q} = \frac{1}{N_q} \sum_{k=1}^{N_q} \mathbf{q}_k \in \mathrm{R}^{D'}$, where $N_q$ is the number of tokens in the question, and $\mathbf{q}_k$ is its $k$-th query vector. The similarity computation and video KV-Cache retrieval are identical to that of external retrieval, except that $\tau$ is set to 1.

Note that, internal retrieval offers several advantages over external retrieval. First, it operates independently within each self-attention layer, allowing different layers to retrieve different video blocks.[2] This allows for a broader capture of video context. Additionally, internal retrieval reuses already computed hidden representations and does not introduce extra parameters, which reduces the computational overhead compared to external retrieval.

**Question-Answering Using Retrieved KV.** The retrieved Video KV-Caches serve as the context for video question-answering. Formally, the attention calculation is formulated as:

$$\mathbf{O} = \mathrm{Attn}\left(\mathbf{W_Q X}, [\mathbf{R}_k, \mathbf{W_K X}], [\mathbf{R}_v, \mathbf{W_V X}]\right), \tag{3}$$

where $\mathbf{X}$ represents either the question tokens or the current token being decoded, and $\mathbf{R}_k$ and $\mathbf{R}_v$ are the key and value vectors from the context, which includes the retrieved video, question, and previously generated tokens.

**Positional Encoding.** Our baseline Video-LLMs employ Rotary Position Embeddings (RoPE) (Su et al., 2024), a commonly used relative positional encoding method. Our video streaming encoding process follows LM-Infinite (Han et al., 2023), where RoPE operates normally within the local window but is constrained by a "distance ceiling" for more distant tokens. For question-answering, we do not account for the original positions of the retrieved KV-Caches, as handling unseen distances among tokens presents significant challenges (Han et al., 2023). Instead, we treat these retrieved tokens as regular consecutive tokens. We also experimented with a static variation from Inf-LLM (Xiao et al., 2024a), where all retrieved tokens are assigned the same position. Our results show that applying standard RoPE to retrieved video tokens leads to better performance, likely due to the importance of capturing temporal information in video comprehension.

## 4 EXPERIMENTS

### 4.1 BENCHMARK AND METRICS

**MLVU$_{\texttt{dev-mc}}$** (Zhou et al., 2024a) is the multiple-choice subset of the MLVU-dev benchmark. It focuses on evaluating the long-form video understanding of MLLMs. The question-answer pairs are manually labeled and can be divided into 3 groups: single-detail, multi-detail, and holistic. The evaluation metric is Accuracy.

**QAEGO4D$_{\texttt{test-mc}}$** (Di & Xie, 2024) is the multiple-choice subset of the QAEGO4D-test benchmark, focusing on question-answering in long egocentric videos. It includes annotations marking video segments relevant to each question. The evaluation metric is Accuracy.

**EgoSchema** (Mangalam et al., 2023) is a diagnostic benchmark for long VideoQA, featuring over 5000 multiple-choice questions and long temporal certificate length. It challenges AI models with long-term understanding, as current state-of-the-art models achieve significantly lower accuracy compared to human performance.

**ActivityNet-QA** (Yu et al., 2019) encompasses human-annotated QA pairs on 5,800 videos derived from the ActivityNet (Caba Heilbron et al., 2015) dataset. This benchmark is designed to assess the

---

[2]For simplicity, we omit the layer index in the above explanation.

capabilities of VideoQA models in long-term spatiotemporal reasoning. Our evaluation methodology aligns with that of Video-ChatGPT (Maaz et al., 2024), employing `GPT-3.5-turbo-0613` to judge the accuracy of the open-ended VideoQA responses.

**RVS-Ego** and **RVS-Movie** (Zhang et al., 2024a) are Streaming VideoQA benchmarks, constructed using 10 long videos from the Ego4D dataset (Grauman et al., 2022) and 22 long videos from the MovieNet dataset (Huang et al., 2020), respectively. These benchmarks feature open-ended questions paired with timestamps, which are initially generated by GPT-4V (OpenAI, 2023b) and GPT-4 (OpenAI, 2023a), and subsequently refined through manual filtering.

Table 1: **Summary of the evaluation benchmarks.** MC stands for multiple-choice VideoQA, while OE refers to open-ended VideoQA.

| Benchmark | Duration | #Videos | #QA | Type |
|---|---|---|---|---|
| MLVU$_{\text{dev-mc}}$ | 12 min | 1,242 | 2,175 | MC |
| QAEGO4D$_{\text{test-mc}}$ | 8.3 min | 148 | 500 | MC |
| EgoSchema | 3 min | 5,031 | 5,031 | MC |
| ActivityNet-QA | 2 min | 800 | 8,000 | OE |
| RVS-Ego | 60 min | 10 | 1,465 | OE |
| RVS-Movie | 30 min | 22 | 1,905 | OE |
| CGBench$_{\text{mc}}$ | 27 min | 1,219 | 12,129 | MC |

**CGBench**$_{\text{mc}}$ (Chen et al., 2025a), the multiple-choice subset of CGBench, is designed for clue-grounded question answering in long videos. It focuses on the ability to retrieve relevant clues for questions, making it an ideal testbed for ReKV.

## 4.2 IMPLEMENTATION DETAILS

We primarily evaluate our approach by integrating it into `LLaVA-OV-0.5B` and `LLaVA-OV-7B` (Li et al., 2024a), chosen for their simplicity and strong performance. In the Appendix, we conduct experiments with several other Video-LLMs as further validations.

All experiments are conducted on NVIDIA A100 (80GB) GPUs with FP16 precision. For video modeling, we process the video stream at 0.5 FPS, in line with GPT-4o's testing on MLVU (Zhou et al., 2024a). The local window size is set to 15K. For external video KV-Cache retrieval, we use `SigLIP-SO400M` (Zhai et al., 2023) as the retriever. For internal KV-Cache retrieval, we set the block size ($b$) to 1 and the number of retrieved frames ($r$) to 64 by default, with further hyper-parameter variations explored in Section 4.3.

Unless otherwise specified, **ReKV** refers to the use of internal video KV-Cache retrieval.

## 4.3 ABLATIONS

In this section, we conduct ablation studies on the effectiveness of in-context retrieval, number of retrieved frames, and the block size.

**Effectiveness of In-context Retrieval.** The experiments on QAEGO4D$_{\text{test-mc}}$, as presented in Table 2, demonstrate the effects of various retrieval methods on VideoQA accuracy and recall. The recall metric, defined as the percentage of question-relevant video frames retrieved, exhibits a strong positive correlation with VideoQA performance: higher recall consistently leads to better accuracy. Uniform Sampling, which sparsely selects frames, achieves the lowest recall and, consequently, the poorest VideoQA accuracy. In contrast, Oracle Retrieval, with perfect recall, delivers the highest VideoQA accuracy, significantly outperforming Uniform Sampling. While External and Internal Retrieval fall short of Oracle-level precision, both surpass Uniform Sampling, with Internal Retrieval excelling due to its higher recall.

Table 2: **Ablation study on QAEGO4D$_{\text{test-mc}}$.** "Oracle Retrieval" refers to a scenario where the annotated, question-relevant video segments are used as input, with a uniform sampling of up to 16 frames. This setup, by definition, has 100% recall and defines the upper-bound VideoQA performance.

| Retrieval Method | VideoQA Acc. | Recall |
|---|---|---|
| `LLaVA-OV-0.5B` | | |
| Uniform Sampling | 42.6 | 6.1 |
| External Retrieval | 48.0 | 58.1 |
| Internal Retrieval | 50.0 | 63.4 |
| Oracle Retrieval | 52.0 | 100 |
| `LLaVA-OV-7B` | | |
| Uniform Sampling | 53.0 | 6.1 |
| External Retrieval | 54.2 | 58.1 |
| Internal Retrieval | 56.0 | 70.5 |
| Oracle Retrieval | 64.4 | 100 |

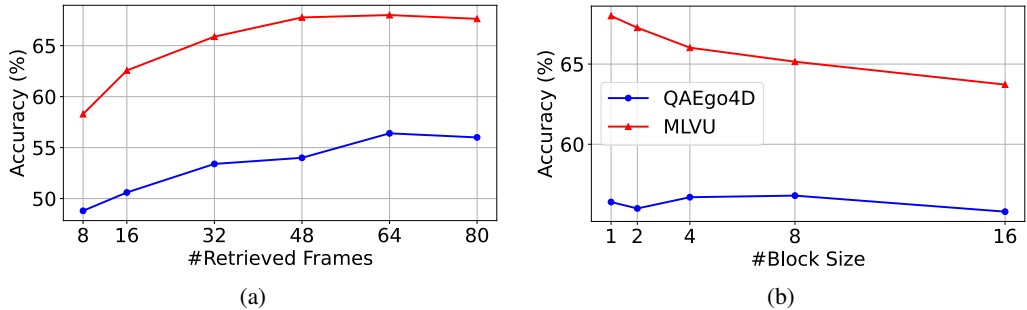

Figure 3: **Ablation study of retrieval hyperparameters:** (a) number of retrieved frames and (b) number of frames per retrieval block. Experiments are conducted with `LLaVA-OV-7B`.

Table 3: **Ablation study on MLVU_dev-mc.** The experiments are based on `LLaVA-OV-7B`.

| Task | Single Detail | | | Multi Detail | | Holistic | | Avg. |
|---|---|---|---|---|---|---|---|---|
| | Needle | Ego | PlotQA | Order | Count | Topic | Anomaly | |
| Uniform Sampling | 74.1 | 59.7 | 69.8 | 45.9 | 32.0 | 87.9 | 72.0 | 64.7 |
| External Retrieval | 78.6 | 69.6 | 71.6 | 40.2 | 37.9 | 84.5 | 63.0 | 66.3 |
| Internal Retrieval | 75.8 | 66.6 | 76.3 | 45.2 | 36.9 | 90.1 | 74.5 | 68.5 |

The MLVU benchmark (Zhou et al., 2024a) encompasses three types of VideoQA tasks: *Single Detail* requires identifying a single critical plot within a long video, *Multi Detail* necessitates the integration of multiple plots, and *Holistic* demands a comprehensive understanding of the entire video. This makes MLVU an ideal platform for evaluating our in-context retrieval method. As shown in Table 3, both External and Internal Retrieval enhance the overall VideoQA accuracy over the Uniform Sampling baseline. The enhancements are most pronounced in Single Detail tasks, demonstrating that ReKV effectively retrieves question-relevant video context. Furthermore, Internal Retrieval significantly outperforms External Retrieval in Holistic tasks, likely due to its ability to capture broader context and leverage the Video-LLM's video modeling capabilities, as discussed in Section 3.

**Number of Retrieved Frames.** We fix the block size ($b = 1$) and evaluate the impact of varying the numbers of retrieved frames ($r \in \{8, 16, 32, 48, 64, 80\}$) on the QAEGO4D and MLVU benchmarks. As illustrated in Figure 3(a), increasing the number of retrieved frames generally improves VideoQA accuracy, as it implies capturing more relevant visual context. However, on MLVU, this improvement plateaus as more frames are retrieved since the additional irrelevant information hinders the subsequent question-answering process. Additionally, retrieving more frames increases the computational overhead of the question-answering stage, further slowing down inference.

**Retrieval Block Size.** When processing video streams, we group $b$ consecutive frames into blocks for block-level retrieval. For this experiment, we fix the number of retrieved frames at $r = 64$ and evaluate different block sizes ($b \in 1, 2, 4, 8, 16$). With a fixed $r$, larger block sizes result in fewer, more concentrated retrieved blocks. Figure 3(b) shows that increasing block size negatively affects accuracy on MLVU, while performance on QAEGO4D remains relatively stable. This suggests that MLVU tasks benefit from retrieving more dispersed visual cues, aligning with its design of multi-detail and holistic tasks (Zhou et al., 2024a). In contrast, QAEGO4D primarily relies on a single relevant clip per question (Di & Xie, 2024).

## 4.4 OFFLINE VIDEO QUESTION-ANSWERING

Streaming video understanding is a relatively under-explored area, with limited StreamingVQA benchmarks available (Zhang et al., 2024a). As discussed in Section 2, OfflineVQA can be considered as a special case of StreamingVQA. Thus, we first evaluate our method in the offline setting using four widely adopted long-form VideoQA benchmarks, comparing our results against state-of-the-art VideoQA methods. A summary of these benchmarks can be found in Table 1.

Table 4: **Offline video question-answering on four long-form benchmarks.** "Acc." denotes accuracy, and "Score" is the open-ended answer rating by `gpt-3.5-turbo-0613` on a scale from 1 to 5.

| Method | MLVU | QAEGO4D | EgoSchema | ActivityNet-QA | |
|---|---|---|---|---|---|
| | dev Acc. | test Acc. | Acc. | Acc. | Score |
| GPT-4V (OpenAI, 2023b) | 49.2 | - | - | 57.0 | - |
| GPT-4o (OpenAI, 2024) | 64.6 | - | - | - | - |
| Gemini-1.5-Flash (Team et al., 2023) | - | - | 65.7 | 55.3 | - |
| Gemini-1.5-Pro (Team et al., 2023) | - | - | 72.2 | 57.5 | - |
| Video-ChatGPT-7B (Maaz et al., 2024) | 31.3 | - | - | - | - |
| LLaMA-VID-7B (Li et al., 2024c) | 33.2 | - | - | 47.4 | 3.30 |
| MiniGPT4-Video-7B (Ataallah et al., 2024) | 44.5 | - | - | 44.3 | 3.35 |
| Video-LLaVA-7B (Lin et al., 2024) | 47.3 | - | - | - | - |
| LongVA-7B (Zhang et al., 2024b) | 56.3 | - | - | 50.0 | - |
| VideoStreaming (Qian et al., 2024) | - | - | 44.1 | - | - |
| Flash-VStream-7B (Zhang et al., 2024a) | 50.2 | 38.2 | 38.1 | 51.9 | 3.40 |
| LLaVA-OV-0.5B (Li et al., 2024a) | 53.2 | 42.6 | 29.6 | 50.5 | 3.02 |
| **+ReKV** (0.5 FPS → 64 Frames) | **56.1** (+2.9) | **50.0** (+7.4) | **31.0** (+1.4) | **52.1** (+1.6) | **3.15** (+.13) |
| LLaVA-OV-7B (Li et al., 2024a) | 64.7 | 52.8 | 59.8 | 56.6 | 3.29 |
| **+ReKV** (0.5 FPS → 64 Frames) | **68.5** (+3.8) | **56.0** (+3.2) | **60.7** (+0.9) | **60.4** (+3.8) | **3.52** (+.23) |

Table 5: **StreamingVQA benchmark results.** All methods are tested under identical conditions. "Video Enc." is frames encoded per second. "Latency" is measured from question input to response completion. "GPU" indicates peak GPU memory usage, and "KV-Cache" refers to the video KV-Cache size offloaded per hour.

| Retrieval Method | RVS-Ego | | RVS-Movie | | Running Speed | | Memory Usage | |
|---|---|---|---|---|---|---|---|---|
| | Acc. | Score | Acc. | Score | Video Enc. | Latency | GPU | KV-Cache |
| Flash-VStream-7B | 57.3 | 4.0 | 53.1 | 3.3 | 14 FPS | 2.4s | 20 GB | - |
| `LLaVA-OV-7B` | | | | | | | | |
| Uniform Sampling | 56.2 | 3.7 | 43.0 | 3.3 | - | 2.9s | 21 GB | - |
| External Retrieval | 62.4 | 3.9 | 53.6 | 3.5 | 11 FPS | 5.8s | 55 GB | 18.8 GB/h |
| Internal Retrieval | 63.7 | 4.0 | 54.4 | 3.6 | 11 FPS | 3.3s | 38 GB | 18.8 GB/h |
| `LLaVA-OV-0.5B` | | | | | | | | |
| Uniform Sampling | 51.8 | 3.7 | 37.2 | 3.2 | - | 2.5s | 7 GB | - |
| External Retrieval | 54.1 | 3.8 | 44.7 | 3.4 | 17 FPS | 4.1s | 37 GB | 4.0 GB/h |
| Internal Retrieval | 54.7 | 3.9 | 44.6 | 3.4 | 17 FPS | 1.6s | 19 GB | 4.0 GB/h |

As shown in Table 4, our proposed ReKV always enhances the performance of `LLaVA-OV-0.5B` and `LLaVA-OV-7B` without additional training. Notably, `LLaVA-OV-7B`+ReKV outperforms two memory-based StreamingVQA models (VideoStreaming (Qian et al., 2024) and Flash-VStream (Zhang et al., 2024a)) by a large margin. While the base model already demonstrates strong performance, and we **do not** claim credit for this achievement, our method can integrate seamlessly with Video-LLMs, benefiting from their ongoing advancements.

## 4.5 STREAMING VIDEO QUESTION-ANSWERING

We then evaluate our method on the streaming setting using the RVS-Ego and RVS-Movie benchmarks. During video stream modeling, questions are input immediately after their annotated end timestamps and answered based on the preceding video content.

**Question-answering Performance.** Table 5 presents the StreamingVQA performance. Both external and internal retrieval methods significantly outperform the uniform sampling baseline. Additionally, our approach enables `LLaVA-OV-7B` to surpass Flash-VStream (Zhang et al., 2024a), demonstrating ReKV's effectiveness for the StreamingVQA.

**Running Speed and Memory Usage.** We also examine the running speed and memory usage under controlled conditions. Specifically, a 1-hour, 1080P video from RVS-Ego with 100 scattered questions is used. Each question is padded to 64 tokens, and the generated answers are fixed at 128 tokens in length. The video frames are pre-extracted at 0.5 FPS (1,800 frames in total) and streamed to the Video-LLM frame by frame.

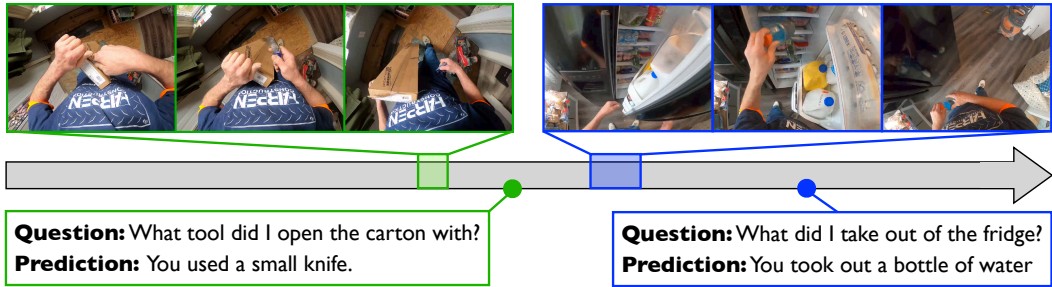

Figure 4: **StreamingVQA qualitative examples.** The example is drawn from the QAEGO4D benchmark. The video stream is processed frame by frame. ● and ● mark the timestamps at which questions are posed. □ and □ indicate the relevant video contexts that support answering these questions.

As illustrated in Table 5, both retrieval methods maintain high video encoding speeds, with `LLaVA-OV-7B` achieving 11 FPS and `LLaVA-OV-0.5B` achieving 17 FPS. Moreover, KV-Cache offloading remains manageable, with `LLaVA-OV-7B` at 18.8GB/h and `LLaVA-OV-0.5B` at 4.0GB/h (see appendix for more details). External retrieval, however, introduces higher latency and GPU memory usage due to additional computations in the external retriever, whereas internal retrieval significantly reduces both. Figure 1 has also demonstrated that latency and GPU memory usage remain stable as more frames are processed. Flash-VStream also shows good efficiency. However, it only maintains a relatively small memory footprint (681 tokens) (Zhang et al., 2024a), leading to potential information loss when dealing with extremely long videos.

**Qualitative Examples.** Figure 4 presents an example of streaming video question-answering. Our approach continuously processes video streams while responding to questions posed at different timestamps. To improve efficiency, it stores and retrieves relevant video KV-Caches as contextual information for answering these questions.

We provide additional implementation details and experimental results in the Appendix.

## 5  RELATED WORK

**LLMs for Video Understanding.** In recent years, there has been a surge of interest in leveraging Large Language Models (LLMs) for video understanding, leading to the development of several innovative approaches (Maaz et al., 2024; Zhang et al., 2024c; Li et al., 2024a). These models typically use a Vision Encoder to extract video features, followed by a mapping step with Linear Projection, MLP, or Q-Former (Li et al., 2023). The mapped features are combined with textual data and fed into large language models (LLMs) to generate a text output. These models have relatively simple architectures, requiring less training data and computational resources, yet they achieve strong performance on short video understanding benchmarks (Xu et al., 2017; Xiao et al., 2021; Li et al., 2024b). However, they employ sparse sampling or token compression techniques to reduce the number of tokens, which can result in significant information loss when dealing with longer or more content-rich videos. As a result, they are not well-suited for long video understanding or streaming video understanding.

**Long Video Understanding.** A central challenge in long video understanding is effectively compressing the information from lengthy videos. Many approaches use language as a bridge, condensing videos into dense captions (Zhang et al., 2023; Islam et al., 2024; Zhou et al., 2024b). While this achieves good results in some cases, compressing video content into text often leads to the loss of crucial visual details. Besides, as a pioneering approach to streaming video understanding, VideoLLM-Online (Chen et al., 2024) employs a data-centric methodology by interleaving video and text during training. In contrast, our approach is training-free, allowing seamless integration with various existing Video-LLMs to extend their StreamingVQA capabilities. Additionally, VideoLLM-Online retains only a single token per frame to handle long videos, which may result in visual information loss. Our method preserves complete visual information and leverages In-Context KV-Cache Retrieval to enhance efficiency.

Another line of research focuses on compressing long videos into a memory bank (Wu et al., 2019; 2022; Wang et al., 2023). MC-ViT (Balazevic et al., 2024) adapts existing pretrained video transformers by fine-tuning them to attend to condensed visual memories. It relates closely to the token-pruning, merging, and memory-based video understanding methods. In comparison, we propose a training-free method specifically tailored to the StreamingVQA task. Incorporating MC-ViT into the StreamingVQA task could be an interesting avenue for future research, and we acknowledge its potential in this domain. This approach has been integrated into Video-LLMs for streaming video understanding, as shown in works like VideoStreaming (Qian et al., 2024) and Flash-VStream (Zhang et al., 2024a). These methods dynamically update the memory during video processing and utilize it for downstream tasks. Despite their innovation, a major limitation of these methods is their failure to account for video length and information density, especially when using a fixed memory size. For example, Flash-VStream compresses both 10-second clips and hour-long videos into the same 681 tokens. Furthermore, these methods lack interpretability, making it difficult to determine how much information is being compressed into the memory or whether relevant video information is being accurately retrieved during downstream tasks.

In pursuit of greater interpretability in long video understanding, methods such as GroundVQA (Di & Xie, 2024) and GeLM (Chen et al., 2025b) advocate for localizing relevant video clips while responding to user queries. Drawing inspiration from these, this work refrains from excessively condensing video information. By harnessing the causal capabilities of Video-LLMs, it preserves the entire Video KV-Cache, allowing for the retrieval of relevant information when required. This strategy effectively mitigates the substantial loss of video content while improving interpretability.

**Long Context Handling for LLMs.** Handling long text sequences in LLMs has been a major challenge due to high computational and memory costs, leading to training constraints on shorter sequences. Techniques like StreamingLLM (Xiao et al., 2024b) and LM-Infinite (Han et al., 2023) use sliding window attention to process long sequences incrementally, but discard distant tokens, limiting the model's ability to capture long-range dependencies. Recent approaches (Xiao et al., 2024a; Li et al., 2025; Fountas et al., 2025) address this by storing and retrieving previously computed KV-Caches, enabling better recall of distant contexts.

**Retrieval-Augmented Generation.** Retrieval-augmented generation (RAG) combines retrieval mechanisms with generative models to enhance performance across various NLP tasks by incorporating external knowledge (Guu et al., 2020; Lewis et al., 2020; Borgeaud et al., 2022) and improving performance in vision-language tasks (Xu et al., 2024). In-context retrieval, recently proposed for handling long inputs (Ram et al., 2023), retrieves information from the input document itself rather than an external knowledge base. In-context KV-Cache retrieval further improves efficiency by pre-encoding long documents, avoiding redundant encodings, and leveraging the LLM's own retrieval capabilities for faster, more effective performance.

## 6 CONCLUSION

In conclusion, this paper introduces a training-free approach, ReKV, designed to enhance the efficiency of Video Large Language Models (Video-LLMs) for streaming video question-answering (StreamingVQA). Unlike conventional video question-answering (VideoQA) systems that must process entire videos before answering, ReKV enables rapid, real-time responses. By employing a sliding-window attention mechanism, it ensures that the model only considers a subset of previous frames while encoding the video stream, significantly cutting down on computational demands. To retain key video context, we developed an in-context KV-Cache retrieval method that efficiently stores and reloads key-value vectors that relevant for each query. This targeted retrieval strategy, combined with the ability to perform video modeling and question-answering on separate processes and GPUs, results in a highly efficient streaming VideoQA system. Extensive experiments show that ReKV not only surpasses existing VideoQA models in performance but also enhances their practicality for real-world streaming applications.

**Acknowledgements.** This work is supported by National Key R&D Program of China (No. 2022ZD0161400). We thank Yikun Liu for discussions and conducting experiments on CGBench.

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
