In the appendix, we provide additional implementation details, experiments, and discussions of limitations and future work.

## A ADDITIONAL IMPLEMENTATION DETAILS

### A.1 MULTI-PROCESSING SERVING

As discussed in Section 2, our approach enables the separation of video modeling and question-answering across different processes and GPUs, significantly enhancing efficiency in real-world applications. Specifically, we dedicate a primary process for video stream encoding, utilizing sliding-window attention to analyze the video and store the computed cache in RAM. If RAM capacity is exceeded, the data can be offloaded to disk. Additionally, a process pool is maintained, with the number of processes determined by the frequency of queries and available resources. Each process loads the same Video-LLM parameters but operates independently. The video processing continues uninterrupted, without waiting for question-answering tasks to complete. When a query is posed, we log its timestamp to ensure that video information after this point is excluded from the answer. An available process from the pool is then activated to retrieve relevant video key-value vectors using our method, loading them onto its GPU for question-answering. This approach enables efficient StreamingVQA applications, with significant potential in areas such as robotics, surveillance, augmented reality, and live broadcasting.

### A.2 PROMPT TEMPLATES FOR VIDEOQA

We use the same prompt template for all multiple-choice VideoQA benchmarks. Text in red indicates variable inputs.

```
System:
You are a helpful assistant.
User:
<video>
Question: <question>
Options:
(A) <Option_A>
(B) <Option_B>
(C) <Option_C>
(D) <Option_D>
(E) <Option_E>
Answer with the option's letter from the given choices directly.
Assistant:
```

The prompt template for open-ended VideoQA is rather simpler:

```
System:
You are a helpful assistant.
User:
<video>
<question>
Assistant:
```

### A.3 KV-CACHE SIZE CALCULATION

The size of the KV-Cache can be calculated using the following formula, assuming FP16 precision:

$$2 \times L \text{ layers} \times T \text{ frames} \times M \text{ tokens/frame} \times H \text{ heads} \times D \text{ dimension} \times 2 \text{ bytes.}$$

For `LLaVA-OV-7B` (Li et al., 2024a), with $L = 28$, $M = 196$, $H = 4$, and $D = 128$, processing a 1-hour video at 0.5 FPS ($T = 1800$) results in a total KV-Cache size of 18.8 GB.

Similarly, for `LLaVA-OV-0.5B` (Li et al., 2024a), with $L = 24$, $M = 196$, $H = 2$, and $D = 64$, processing a 1-hour video at 0.5 FPS results in a total KV-Cache size of 4.0 GB.

These theoretical calculations are consistent with the experimental results shown in Table 5.

# B ADDITIONAL EXPERIMENTS

## B.1 EXPERIMENTS WITH MORE VIDEO-LLMS AND BENCHMARK

To further assess the generalizability of our approach, we tested it on additional Video-LLMs: `Video-LLaVA-7B` (Lin et al., 2024), `LongVA-7B` (Zhang et al., 2024b), and `LLaVA-OV-72B` (Li et al., 2024a). We used model sharding for `LLaVA-OV-72B`, significantly slowing inference. To mitigate this, we reduced the FPS to 0.1 and the number of retrieved frames to 32, ensuring efficient evaluation. As shown in Table 6, ReKV consistently improved performance across various models and benchmarks, highlighting its robustness and adaptability.

Table 6: **Additional experiments with more Video-LLMs and benchmark.**
"Acc." denotes accuracy. "X Frames" refers to uniformly sampling X frames from the video. "Y FPS → X Frames" indicates an input video with a frame rate of Y FPS, from which X frames are retrieved.

| Method | Sampling | MLVU | QAEGO4D | EgoSchema | CGBench |
|---|---|---|---|---|---|
| | | dev Acc. | test Acc. | Acc. | Acc. |
| Video-LLaVA-7B (Lin et al., 2024) | 8 Frames | 46.5 | 37.0 | 41.4 | 18.7 |
| **+ReKV** | 0.5 FPS → 8 Frames | **47.2** (+0.7) | **37.9** (+0.9) | **42.2** (+0.8) | **19.2** (+0.5) |
| LongVA-7B (Zhang et al., 2024b) | 32 Frames | 57.3 | 42.8 | 42.5 | 26.1 |
| **+ReKV** | 0.5 FPS → 32 Frames | **58.6** (+1.3) | **45.6** (+2.8) | **42.7** (+0.2) | **26.4** (+0.3) |
| LLaVA-OV-0.5B (Li et al., 2024a) | 64 Frames | 53.2 | 42.6 | 29.6 | 21.4 |
| **+ReKV** | 0.5 FPS → 64 Frames | **56.1** (+2.9) | **50.0** (+7.4) | **31.0** (+1.4) | **21.7** (+0.3) |
| LLaVA-OV-7B (Li et al., 2024a) | 64 Frames | 64.7 | 52.8 | 59.8 | 31.1 |
| **+ReKV** | 0.5 FPS → 64 Frames | **68.5** (+3.8) | **56.0** (+3.2) | **60.7** (+0.9) | **33.9** (+2.8) |
| LLaVA-OV-72B (Li et al., 2024a) | 32 Frames | 71.9 | 53.6 | 59.6 | 37.2 |
| **+ReKV** | 0.1 FPS → 32 Frames | **72.6** (+0.7) | **57.0** (+3.4) | **62.0** (+2.4) | **40.5** (+3.3) |

## B.2 FAIR COMPARISONS WITH FLASH-VSTREAM

Tables 4 and 5 compared `LLaVA-OneVision+ReKV` with `Flash-VStream`. However, these comparisons may be unfair due to different architecture and training data. Thus, here we conduct **fair** comparisons using the same Video-LLM backbone, including the identical visual encoder (`CLIP-ViT-L/14`), projector (2-layer MLP), LLM (`Vicuna-7B-v1.5`), training data, and train/eval pipelines.

Due to the inaccessibility of WebVid videos[3] used in Flash-VStream's original training, we use 232K randomly sampled InternVid videos[4] as a substitute. This ensures comparable experimental settings. We train a baseline Video-LLM model (`Base`) and a Flash-VStream-enhanced version (`Base+Flash`). Similarly, we integrate ReKV into the same baseline (`Base+ReKV`) for fair comparisons. To maintain parity, the baseline uniformly samples 16 frames per video, resized to $224 \times 224$. Visual features of shape $(T, 16, 16, D)$ are average-pooled to $(T, 8, 8, D)$ before being passed through the MLP projector and into the LLM. Both Flash-VStream and ReKV process video at 0.5 FPS, with ReKV retrieving 16 frames.

Table 7: **Fair comparisons with Flash-VStream.** "Original Flash" is the checkpoint officially published by Flash-VStream while "Base+Flash" is our reproduced version.

| Method | MLVU$_{dev-mc}$ | QAEGO4D$_{test-mc}$ | EgoSchema | RVS-Movie | RVS-Ego |
|---|---|---|---|---|---|
| Base | 49.8 | 39.0 | 42.6 | 47.2 | 54.1 |
| Base+Flash | 51.0 | 37.4 | 41.2 | 50.1 | **55.4** |
| **Base+ReKV** | **51.9** (+0.9) | **40.5** (+3.1) | **43.7** (+2.5) | **51.9** (+1.8) | 54.7 (-0.7) |
| Original Flash | 50.2 | 38.2 | 38.1 | 53.1 | 57.3 |

As shown in Table 7, `Base+ReKV` consistently outperforms the base Video-LLM `Base` and surpasses `Base+Flash` in most cases, highlighting its superiority under fair comparative conditions. Additionally, ReKV offers enhanced usability, seamlessly integrating with existing Video-LLMs without requiring extensive retraining.

---

[3]https://github.com/m-bain/webvid
[4]https://huggingface.co/datasets/OpenGVLab/InternVid

On the contrary, the reproduced `Base+Flash` does not consistently outperform `Base`. It excels on StreamingVQA (RVS-Movie and RVS-Ego) and MLVU but underperforms on QAEGO4D and EgoSchema. This discrepancy is likely due to significant visual information loss: the `Base` model processes 1024 visual tokens ($16 \times 64$), while `Base+Flash` uses only 681 memory tokens.

For additional context, we include results from the original Flash-VStream (`Original Flash`) using checkpoints from its official repository[5]. Our reproduced `Base+Flash` shows performance deviations, likely due to differences in training data and potential environmental factors.

### B.3 COMPUTATIONAL COMPLEXITY

We ensure **fair** comparisons by using the identical Video-LLM backbone (Sec. B.2) under controlled streaming conditions (Sec. 4.5). Specifically, we measured the FLOPs and MACs of the base Video-LLM, Flash-VStream, and our external and internal retrieval methods. We analyzed **average TFLOPs and TMACs per QA over various question frequencies** in a 1-hour video, leveraging the `calflops` library (Ye, 2023).

As shown in Tables 8 and 9, ReKV's efficiency improves significantly with increasing QA frequency. The video stream is encoded only once, and computed results are reused across QAs, leading to reduced per-query complexity as QA frequency rises. Flash-VStream outperforms ReKV at low QA frequencies (*e.g.*, 100 QAs). However, ReKV's complexity decreases more rapidly with increased QA frequency, primarily due to Flash-VStream's high memory update overhead. ReKV is thus better suited for high-concurrency scenarios such as live streaming and requires no additional training.

Furthermore, Internal retrieval consistently outperforms external retrieval, reducing average FLOPs by 15.5% and MACs by 15.2%. These results underscore ReKV's ability to balance computational efficiency and effectiveness, particularly in dynamic, high-query environments. This positions ReKV as a practical and scalable solution for streaming video understanding.

Table 8: **TFLOPs / QA.**

| #QAs | Baseline | Flash-VStream | ReKV (External) | ReKV (Internal) |
|---|---|---|---|---|
| 100 | 22.4 | **15.5** | 21.7 | 18.5 |
| 200 | 12.7 | 14.1 | 11.4 | **9.6** |
| 360 | 8.5 | 13.8 | 6.8 | **5.6** |

Table 9: **TMACs / QA.**

| #QAs | Baseline | Flash-VStream | ReKV (External) | ReKV (Internal) |
|---|---|---|---|---|
| 100 | 11.2 | **7.8** | 10.8 | 9.2 |
| 200 | 6.4 | 7.1 | 5.7 | **4.8** |
| 360 | 4.3 | 6.8 | 3.3 | **2.8** |

## C LIMITATIONS AND FUTURE WORK

While ReKV improves the accuracy and efficiency of Video-LLMs in the StreamingVQA task, it still has several limitations that deserves future investigation: *First*, although the KV-Cache offloading to RAM or disk is manageable, as shown in Table 5, handling extremely long video streams, such as those in surveillance, may lead to an unsustainable increase in cache size. This issue can be mitigated by integrating techniques such as quantization, token pruning, and compression. *Second*, the use of a constant block size for grouping consecutive frames during retrieval can disrupt video continuity. A more refined solution would involve segmenting videos into semantically coherent blocks. *Third*, our method retrieves a fixed number of frames. Future work could explore dynamic retrieval strategies that adjust the number of frames based on video context and query requirements. *Finally*, StreamingVQA remains an under-explored task with few available benchmarks. Developing high-quality benchmarks with precise temporal annotations is crucial for advancing future research.

---

[5]https://github.com/IVGSZ/Flash-VStream