# OpenReview forum: "Streaming Video Question-Answering with In-context Video KV-Cache Retrieval"
_ICLR.cc/2025/Conference — ICLR 2025 Poster_

### Official Review · Reviewer_3gu5 · 2024-10-30

**Soundness:** 4
**Presentation:** 2
**Contribution:** 3
**Rating:** 6
**Confidence:** 4

**Summary:**

The paper introduces ReKV, a novel, training-free approach designed to enhance the efficiency of Video Large Language Models (Video-LLMs) for streaming video question-answering (StreamingVQA). Unlike traditional VideoQA systems that process entire videos before answering queries, ReKV processes video streams in real-time, allowing for prompt responses. The method employs a sliding-window attention mechanism to reduce computational overhead and uses a KV-Cache system to store and retrieve relevant video information efficiently. The approach separates video encoding and question-answering into distinct processes, enhancing efficiency. The paper demonstrates the efficacy of ReKV through comprehensive experiments, showing improvements in accuracy, latency, and memory usage over existing models.

**Strengths:**

1.	Efficiency: The sliding-window attention mechanism and KV-Cache retrieval significantly reduce computational overhead and memory usage.
2.	Real-Time Processing: The method allows for real-time responses to queries, making it highly practical for applications like surveillance and live broadcasts.
3.	Comprehensive Evaluation: The paper provides extensive experimental results, demonstrating the effectiveness of ReKV across multiple benchmarks.
4.	Seamless Integration: ReKV integrates seamlessly with existing Video-LLMs without requiring additional training, making it easy to adopt.

**Weaknesses:**

1.	Writing Quality: The organization of this paper could be improved. It is not appropriate to place ablation study before main experiments. Sec 2.1 Task definition and discussion should not be a part of Method. This part is too repetitive of the discussion in the introduction.
2.	Citation format: In Table 4 Line 391 there may be a misleading citation of Video-LLaVA-7B, pointing to the same reference of Video-ChatGPT-7B.
3.	Lack explanation: The term "oracle retrieval" from Table 2 and Line 305 is difficult for readers to understand. How is the “recall” metric calculated? How can it be 100?

**Questions:**

1.	Generalizability of method: Since ReKV is a training-free method, can it be integrated with models other than LLaVA-OV? Are there any experimental results?
2.	Scalability: How does ReKV scale with increasing video length and complexity? Are there any observed limitations when dealing with very high-resolution videos or videos with a high frame rate?
3.	Implementation details: What is the hyperparameters of external retrieval?

---

> ### Author Response · Authors · 2024-11-24
> **Response to Reviewer 3gu5 (Part1)**
>
> Thank you for your constructive comments. We have provided our responses below.
>
> ---
>
> **Q1: Paper Organization**
>
> We greatly appreciate your thoughtful feedback and suggestions to enhance the organization and clarity of our paper. Specifically:
>
> - **Placement of Ablation Study:** We placed the ablation study early to immediately demonstrate the core advantage of our ReKV method over uniform sampling. This choice allows us to establish its effectiveness upfront, followed by broader experimental results. Tables 4-5 further demonstrate that our training-free, easily integrable approach achieves SOTA performance with VideoLLMs.
> - **Separating Sec 2.1 and 2.2:** We agree with the suggestion and will revise our draft accordingly.
> - **Necessity of Sec 2.1:** StreamingVQA is a relatively new research area with varying perspectives in prior works [1-3]. While the introduction provides a high-level overview, we believe it is important to formally define the task and discuss design principles in the main body to establish a solid foundation for our methodology.
>
> We will incorporate these changes to ensure better structure and coherence in the final version. Thank you again for your valuable feedback.
>
> [1] Chen, Joya, et al. "VideoLLM-online: Online Video Large Language Model for Streaming Video." In CVPR. 2024.
>
> [2] Qian, Rui, et al. "Streaming long video understanding with large language models." arXiv:2405.16009, 2024.
>
> [3] Zhang, Haoji, et al. "Flash-VStream: Memory-Based Real-Time Understanding for Long Video Streams." arXiv:2406.08085, 2024.
>
> ---
>
> **Q2: Citation Error**
>
> Thank you for pointing this out. The citation error will be corrected in the revised manuscript.
>
> ---
>
> **Q3: Clarifications for Table 2**
>
> We appreciate the reviewer’s feedback regarding the term “Oracle Retrieval” and the calculation of the “recall” metric. To clarify:
>
> - The QAEgo4D dataset includes annotations marking video segments relevant to each question. For example, for the question, `Where did I put the dog fur?`, the dataset provides not only the answer on the table but also a temporal window `4-6 seconds` indicating the video segment leading to the answer.
> - **Oracle Retrieval** in Table 2 refers to a scenario where these annotated, question-relevant video segments are directly used as input, bypassing the retrieval process. This setup defines the upper-bound performance.
> - The **Recall** metric is defined as the percentage of question-relevant video frames retrieved. Since the “Oracle Retrieval” scenario directly utilizes annotations to identify relevant segments, the recall is 100% by definition.
>
> We will include this explanation in the revised manuscript to ensure clarity for readers.

---

> ### Author Response · Authors · 2024-11-24
> **Response to Reviewer 3gu5 (Part2)**
>
> **Q4: Generalizability**
>
> We kindly refer you to our *General Response*, where we present experiments with various Video-LLMs to address this concern.
>
> ---
>
> **Q5: Scalability**
>
> Thanks for your question regarding scalability. We address this across several dimensions:
>
> - **Video length:** ReKV scales effectively with varying video lengths. As illustrated in Figure 1b, ReKV consistently outperforms the Uniform Sampling baseline across six benchmarks, regardless of video length.
> - **Number of retrieved frames:** Performance improves with an increasing number of retrieved frames, as shown in Figure 3a (ranging from 8 to 64 frames). This performance gain saturates beyond 64 frames, primarily due to the base Video-LLM’s limitations (e.g., LLaVA-OV, trained on a maximum of 32 frames, struggles to effectively process a larger number of retrieved frames).
> - **Model Complexity:** ReKV adapts seamlessly to models of various sizes. Our evaluations on LLaVA-OV (ranging from 0.5B to 72B parameters) and other Video-LLMs demonstrate its scalability across model complexities.
> - **Video Resolution:** Scalability with video resolution depends on the base Video-LLM, which typically resizes frames to fixed dimensions (e.g., 384x384 for LLaVA-OV and 224x224 for Video-LLaVA). Increased resolution primarily impacts the size of the KV-Caches rather than ReKV’s performance.
> - **Frame Rate (FPS):** As shown below, ReKV achieves optimal performance around 0.5-1 FPS. Lower FPS degrades performance due to significant visual information loss, while higher FPS adds excessive irrelevant context, potentially distracting the retrieval process.
>
> Experiments on the QAEgo4D test set (retrieve 64 frames).
> | FPS             | 5    | 2    | 1    | 0.5  | 0.2  | 0.1  |
> |------------------|------|------|------|------|------|------|
> | LLaVA-OV-7B + **ReKV** | 52.5 | 53.8 | 56.2 | **56.4** | 52.4 | 51.2 |
>
> ---
>
> **Q6: Hyperparameters of External Retrieval**
>
> We maintained identical hyperparameters for external and internal retrieval to ensure a fair comparison. Specifically, we set block size $b = 1$ and the number of retrieved frames $r = 64$ for both methods (L287).

---

> ### Comment · Reviewer_3gu5 · 2024-11-26
>
> Thank you for your detailed rebuttal. Your responses have satisfactorily addressed my concerns. I believe your paper is now much improved.

---

> > ### Author Response · Authors · 2024-11-26
> >
> > Thank you for your time and positive feedback. We will carefully review your suggestions and incorporate them into a revised version.

---

### Official Review · Reviewer_g3YS · 2024-11-01

**Soundness:** 3
**Presentation:** 2
**Contribution:** 2
**Rating:** 6
**Confidence:** 4

**Summary:**

The authors present ReKV (Retrieve In-context Video KV-Cache) for streaming video question-answering. The authors incorporate a sliding-window attention mechanism on existing VideoLLMs, introduce a retrieval method that leverages an external retriever or the parameters within Video-LLMs to retrieve only queryrelevant KV-Caches. The authors evaluates the model on both long video QA and streaming videoqa.

**Strengths:**

1. The number of evaluation benchmark for the proposed method is adequate.

2. The improvements against standard VideoLLMs are substantial.

**Weaknesses:**

1. Lack of fair comparison against existing memory-based models (including VideoStreaming and Flash-VStream). It would be better if the author could provide results for ReKV and previous memory-based models under the same VideoLLM backbone to show the effectiveness of the proposed method, for both long video benchmarks and streaming video benchmarks.

2. Missing related works. The authors should discuss the novel contribution compared to the paper VideoLLM-online[1], MC-ViT[2].

3. The “External Video KV-Cache Retrieval” is confusing, do the authors mean selecting the keyframes using the query information via the CLIP-based models (like a cross-modal matching)? This is already investigated by a number of works, including ATP [3], SeViLA[4] and so on.  It would be if for the authors to clarify how "External Video KV-Cache Retrieval" differs from or improves upon the keyframe selection methods.

[1] VideoLLM-online: Online Video Large Language Model for Streaming Video

[2] Memory Consolidation Enables Long-Context Video Understanding

[3] Revisiting the "Video" in Video-Language Understanding

[4] Self-Chained Image-Language Model for Video Localization and Question Answering

**Questions:**

All weakness, and:

1. The citation format is inconsistent over the paper, the authors should unify this format.

2. Since the model is claimed to integrate seamlessly with existing Video-LLMs, is it possible to apply to a bigger VideoLLM backbone, like around 70B scale?

3. In Table 4,  for offline video question-answering, could the authors elaborate more on the baseline setting of the LLaVA-ov model, like frame numbers? Also, could the authors compare the efficiency of the proposed ReKV compared to the original LLaVA-ov model in the table?

---

> ### Author Response · Authors · 2024-11-24
> **Response to Reviewer g3YS (Part1)**
>
> Thanks for your constructive comments. We provide our responses as follows.
>
> ---
>
> **Q1: Missing related works**
>
> Thank you for the suggestion regarding related works.
>
> - As a pioneering approach to streaming video understanding, VideoLLM-Online employs a data-centric methodology by interleaving video and text during training. In contrast, our approach is training-free, allowing seamless integration with various existing Video-LLMs to extend their StreamingVQA capabilities. Additionally, VideoLLM-Online retains only a single token per frame to handle long videos, which may result in visual information loss. Our method preserves complete visual information and leverages In-Context KV-Cache Retrieval to enhance efficiency.
> - MC-ViT adapts existing pretrained video transformers by fine-tuning them to attend to condensed visual memories. It relates closely to the token-pruning, merging, and memory-based video understanding methods. In comparison, we propose a training-free method specifically tailored to the StreamingVQA task. Incorporating MC-ViT into the StreamingVQA task could be an interesting avenue for future research, and we acknowledge its potential in this domain.
>
> We will incorporate detailed discussions on these comparisons in our revised draft to clarify the novelty and contributions of our work. Thank you again for pointing this out.
>
> ---
>
> **Q2: Clarification of External Retrieval**
>
> External retrieval is a straightforward cross-modal retrieval approach using a CLIP-like model (`SigLIP-SO400M` in our implementation) and is not positioned as a core contribution. Instead, it serves as a training-free baseline to help validate the effectiveness of our In-Context KV-Cache Retrieval framework. While existing keyframe selection or moment retrieval methods can also identify query-relevant video frames, they typically require additional training, making them incompatible with our training-free framework.
>
> ---
>
> **Q3: Citation Format Issues**
>
> Thank you for pointing out the citation inconsistencies. We will address and correct them in the revised manuscript.
>
> ---
>
> **Q4: Can ReKV Work with Bigger VideoLLMs?**
>
> Yes, ReKV scales to larger VideoLLMs. Using [Accelerate](https://huggingface.co/docs/accelerate/en/concept_guides/big_model_inference), we distribute model layers across multiple GPUs, ensuring proper placement of inputs on each GPU. Attention calculations remain as outlined in our paper. Experiments with `LLaVA-OV-72B` in our *General Response* confirm that ReKV significantly improves performance.

---

> ### Author Response · Authors · 2024-11-24
> **Response to Reviewer g3YS (Part2)**
>
> **Q5: Frame Numbers and Efficiency for OfflineVQA**
>
> Thank you for your valuable suggestions. We have conducted additional experiments to address your concerns. Specifically, we varied the number of input (or retrieved) frames and reported QAEgo4D accuracy for each configuration. To address efficiency concerns, we split the VideoQA process into video encoding and question-answering (L150) and measured the average processing time per QA using an NVIDIA H800 GPU.
>
> |                | # Frames | QAEgo4D Acc. | Video Enc. (s) | QA (s) |
> |----------------|----------|--------------|----------------|--------|
> | LLaVA-OV-7B    | 8        | 48.4         | 5.0            | 0.1    |
> |                | 16       | 49.6         | 5.2            | 0.1    |
> |                | 32       | 51.2         | 5.2            | 0.2    |
> |                | 64       | 50.8         | 5.4            | 0.3    |
> | &ensp;**+ReKV**          | 8        | 49.2         | 41.5           | 0.1    |
> |                | 16       | 50.8         | 41.3           | 0.1    |
> |                | 32       | 53.4         | 41.3           | 0.2    |
> |                | 64       | 56.4         | 41.5           | 0.3    |
>
> Our findings indicate that:
>
> - Both LLaVA-OV and LLaVA-OV + ReKV improve performance as the number of frames increases.
> - ReKV consistently outperforms baseline methods, with performance gains increasing as more frames are added.
> - ReKV primarily adds inference time to the video encoding process due to encoding substantially more frames, while the QA process remains highly efficient. Notably, our method is designed for the StreamingVQA setting, where video encoding continuously processes frames (11 FPS in our experiments, as shown in Table 5). ReKV demonstrates strong efficiency under these conditions, as evidenced in Table 5 and our *Response to Reviewer AcRC (Part2)*.
>
> We will incorporate the analysis, along with additional benchmarks and model comparisons, in our final draft.

---

> ### Author Response · Authors · 2024-11-25
> **Response to Reviewer g3YS (Part3)**
>
> **Q6: Fair comparisons with FlashVStream and VideoStreaming**
>
> We appreciate the reviewer’s concern regarding fair comparisons. Below, we address the points raised:
>
> - **Comparisons with Flash-VStream.**
> | Model            | MLVU dev | QAEgo4D test | EgoSchema | RVS-Movie | RVS-Ego  |
> | ---------------- | -------- | ------------ | --------- | --------- | -------- |
> | Base             | 49.8     | 39.0         | 42.6      | 47.2      | 54.1     |
> | Base+Flash       | 51.0     | 37.4         | 41.2      | 50.1  | **55.4**     |
> | **Base+ReKV**    | **51.9** | **40.5**     | **43.7**  | **51.9**    | 54.7 |
> | *Original Flash* | *50.2*   | *38.2*       | *38.1*    | *53.1*    | *57.3*   |
>   - We conducted fair comparisons between Flash-VStream and our proposed ReKV using the same Video-LLM backbone, including the identical visual encoder (CLIP-ViT-L/14), projector (2-layer MLP), LLM (Vicuna-7B-v1.5), training data, and train/eval pipelines.
>   - **Implementation Details:**
>     - Due to the inaccessibility of WebVid videos [1] used in Flash-VStream’s original training, we used 232K randomly sampled InternVid videos [2] as a substitute. This ensured comparable experimental settings.
>     - We trained a baseline Video-LLM model (`Base`) and a Flash-VStream-enhanced version (`Base+Flash`). Similarly, we integrated ReKV into the same baseline (`Base+ReKV`) for a direct comparison.
>     - To maintain parity, the baseline processes uniformly sampled $16$ frames per video, resized to $224\times224$. Visual features ($T,16,16,D$) are average-pooled to $(T,8,8,D)$ before being passed through the MLP projector and into the LLM. Both Flash-VStream and ReKV process video at 0.5 FPS, with ReKV retrieving 16 frames.
>   - **Analysis:**
>     - **ReKV:** `Base+ReKV` **consistently outperforms** the base Video-LLM  `Base` and **surpasses** `Base+Flash` in most cases, highlighting its superiority under fair comparative conditions. Additionally, ReKV offers **enhanced usability**, seamlessly integrating with existing Video-LLMs without requiring extensive retraining.
>     - **Flash-VStream:** The reproduced `Base+Flash` does not consistently outperform `Base`. It excels on StreamingVQA (RVS-Movie and RVS-Ego) and MLVU but underperforms on QAEgo4D and EgoSchema. This discrepancy is likely due to significant visual information loss: the `Base` model processes 1024 visual tokens ($16 \times 64$), while `Base+Flash` uses only 681 memory tokens.
>     - **Reproduction:** For additional context, we include results from the original Flash-VStream (`Original Flash`) using checkpoints from its official repository [3]. Our reproduced `Base+Flash` shows performance deviations, likely due to differences in training data and potential environmental factors.
>
> - **Comparisons with VideoStreaming.**
>   - Direct comparisons are infeasible since VideoStreaming has not been open-sourced.
>   - Moreover, it employs a specialized architecture with an additional LLM (`Phi-2-2.7B`) as a streaming encoder, incorporating additional parameters. This architectural divergence complicates fair, apples-to-apples comparisons.
>
> We will incorporate these analyses into our final draft. Thanks again for your valuable suggestion.
>
> ---
>
> [1] https://github.com/m-bain/webvid
>
> [2] https://huggingface.co/datasets/OpenGVLab/InternVid
>
> [3] https://github.com/IVGSZ/Flash-VStream

---

> > ### Comment · Reviewer_g3YS · 2024-11-27
> >
> > Thanks the authors for the detailed response, I would like to see the updated results in the final draft. I am happy to raise my score to 6.

---

> > > ### Author Response · Authors · 2024-11-28
> > >
> > > Thank you for your kind feedback and for raising the score!

---

> ### Author Response · Authors · 2024-11-26
> **Gentle Reminder for Your Feedback**
>
> Dear Reviewer `g3YS`,
>
> Thank you for your time and effort in reviewing our submission. We have carefully considered your comments and provided detailed responses. We look forward to your feedback.

---

### Official Review · Reviewer_t8DB · 2024-11-03

**Soundness:** 4
**Presentation:** 4
**Contribution:** 4
**Rating:** 8
**Confidence:** 4

**Summary:**

This paper introduces ReKV, a novel, training-free approach designed to enhance existing Video-LLMs for StreamingVQA. Traditional VideoQA systems struggle with long videos due to the need to process entire videos before responding and repeating this process for each new question. ReKV addresses these challenges by storing processed video key-value caches (KV-Caches) in RAM or disk to prevent information loss. ReKV introduces retrieval methods—both external (using models like CLIP) and internal (leveraging the Video-LLM's parameters)—to fetch only query-relevant KV-Caches, enhancing efficiency and accuracy in question-answering.
Experiments conducted on various benchmarks, including MLVU, QAEGO4DMC, EgoSchema, ActivityNet-QA, and StreamingVQA (RSV-Ego and RSV-Movie) datasets, demonstrate that ReKV improves VideoQA accuracy while maintaining stable inference latency and memory usage as the number of frames increases. The method enables real-time interaction and long-term context for StreamingVQA tasks.

**Strengths:**

- The paper presents a novel and simple, training-free method that extends the capabilities of existing Video-LLMs for StreamingVQA. By integrating a sliding-window attention mechanism and efficient KV-Cache retrieval, ReKV addresses the challenges of processing long video streams in real-time.

- The methodology is well-motivated and thoroughly explained. The paper clearly defines the StreamingVQA task, differentiates it from traditional OfflineVQA, and outlines the specific challenges involved. The proposed solutions are detailed and logically sound.

-  The paper is well-organized and clearly written with figures to support the method explanation.

- ReKV significantly improves efficiency and accuracy over existing VideoQA models on multiple benchmarks. The ability to handle long video streams in a streaming fashion has practical importance for real-world applications. The training-free nature of ReKV can potentially enhance its applicability across different Video-LLMs.

**Weaknesses:**

Currently, a major limitation of the method is that the method is that it is only evaluated on LLaVA-OV models (0.5B and 7B). Although these models are strong baselines, the applicability of ReKV to other Video-LLMs is not demonstrated. Evaluating ReKV on a broader set of models would strengthen the claim of its versatility and general applicability.
I’ll be happy to increase my score if that limitation is addressed.

**Questions:**

- Have the authors tested ReKV with other Video-LLMs besides LLaVA-OV? Demonstrating the integration and performance of ReKV with different architectures (e.g., VideoChatGPT…) would confirm its general applicability and ease of integration.
- Table 5 shows that the internal KV-Cache retrieval reduces computational overhead compared to external retrieval. However the “internal retrieval” retrieves KV-Caches for each attention layer independently while it is only done once for the “external retrieval”. How do you explain that the internal is faster?


Minor:
In practice, how does ReKV manage KV-Cache storage for extremely long video streams, such as surveillance footage that can run continuously for many hours or days? Are there mechanisms in place to prevent unsustainable increases in cache size, and how does this impact performance and resource requirements?

---

> ### Author Response · Authors · 2024-11-24
> **Response to Reviewer t8DB**
>
> Thanks for your positive comments! We provide our feedback as follows.
>
> ---
>
> **Q1: Evaluation of ReKV on a Broader Range of Models**
>
> We kindly refer you to our *General Response*, where we present experiments with various Video-LLMs to address this concern.
>
> ---
>
> **Q2: Why does Internal Retrieval reduce computational overhead over External Retrieval?**
>
> While internal retrieval operates at every layer, it efficiently reuses the LLM KV-Caches and performs fast cosine similarity calculations. In contrast, external retrieval incurs higher overhead due to the need for an additional retriever to encode frames and questions, making it more computationally expensive overall.
>
> As detailed in our *Response to Reviewer AcRC (Part2)*, internal retrieval achieves a **15.5% reduction in average FLOPs** and a **15.2% reduction in MACs**, highlighting its superior efficiency over external retrieval.
>
> ---
>
> **Q3: How does ReKV manage KV-Cache storage?**
>
> ReKV offloads KV-Caches from GPU to RAM and further to disk when RAM capacity is exceeded (Appendix A.1). Table 5 illustrates that `LLaVA-OV-7B` produces 18.8 GB of KV-Caches for an hour-long video, scaling to 450 GB for a day-long video. This size is manageable for modern surveillance systems.
>
> Section 6 discusses recent advancements in reducing KV-Cache sizes, such as quantization, token pruning, and compression, which are complementary to our method. If their methods do not harm performance on their own, integrating them with ours would not degrade performance either.

---

> > ### Comment · Reviewer_t8DB · 2024-11-25
> >
> > Thank you for addressing my main concern regarding ReKV's applicability to different Video-LLMs. The additional experiments with Video-LLaVA-7B, LongVA-7B, and LLaVA-OV-72B demonstrate the method's broad applicability and effectiveness. Given these new results, I am increasing my evaluation score.

---

> > > ### Author Response · Authors · 2024-11-26
> > >
> > > Thank you for your positive feedback and thoughtful evaluation. We truly appreciate your kind words.
> > >
> > > However, it seems the review score has not yet been updated. We would be grateful if you could kindly increase the score. Thank you again for your support!

---

### Official Review · Reviewer_AcRC · 2024-11-03

**Soundness:** 2
**Presentation:** 3
**Contribution:** 1
**Rating:** 6
**Confidence:** 3

**Summary:**

This paper presents video KV caches to make a streaming video question and answering video-LLMs in a training-free approach.
While it uses a sliding-attention mechanism to aggregate short-term temporal context, video KV caches and the proposed retrieval method are introduced to long-term temporal context.
This method shows efficiency with LLaVA-OV in several benchmarks.

**Strengths:**

- This paper outperforms existing video-LLMs on long-form benchmarks.
- This paper is easy to follow.
- Ablation study shows the validity and impact of the retrieval methods.

**Weaknesses:**

- My major concern is the novelty. Already, many LLM systems reduce the context-processing delay by using the KV cache of the context. This paper is also built on LLMs, while it is coupled with a video encoder. It's hard to find the specialty for the video streaming system. The causal attention and retrieval system with cosine similarity are also not new.

- Implementation with only one design (LLaVA-OV) with different sizes is limited to prove the generality of the proposed method.

- There are many recent methods to reduce the memory of KV caches such as adaptive KV cache (ICLR'24) and Keyformer (Muhammad Adnan et al., arxiv'24), compared to these methods, is the proposed attention and search method more effective?

**Questions:**

- How about GFLOPs on streaming VQA?

---

> ### Author Response · Authors · 2024-11-24
> **Response to Reviewer AcRC (Part1)**
>
> Thank you for your constructive comments. Our responses are provided below.
>
> ---
>
> **Q1: Clarifying the Novelties**
>
> Thank you for your feedback. We appreciate the opportunity to clarify the novelty and contributions of our work.
>
> (1) The core novelty of our work lies in the formal definition and discussion of the StreamingVQA task, a relatively under-explored domain with broad real-world applications (L37). We highlight that OfflineVQA is a special case of StreamingVQA. Existing methods, however, suffer from substantial visual information loss and inefficiency caused by repeated computations. To bridge these gaps, we propose In-context Video KV-Cache Retrieval for efficient and scalable StreamingVQA, introducing a fresh perspective absent in prior research on video understanding and MLLMs.
>
> (2) StreamingVQA presents distinct challenges, such as long-context handling and cross-modal retrieval in high-dimensional, redundant video data. While recent advances in LLMs inform our approach (as discussed in Related Work, L498-511), our contributions focus on adapting and extending these techniques to the streaming video domain, including sliding-window video encoding, video KV-cache offloading, and internal video KV-Cache retrieval. Notably, our simple, training-free method integrates seamlessly with existing Video-LLMs for StreamingVQA, a feature appreciated by Reviewer t8DB and 3gu5.
>
> (3) Influential MLLM works (e.g., LLaVA [1] and LongVA [2]) demonstrate the value of leveraging LLM advancements to address domain-specific challenges. For instance, LLaVA applies instruction tuning [3] to multimodal tasks, while LongVA transfers long-context capabilities [4] to MLLMs. Similarly, our work pushes the boundaries by extending long-context handling and cross-modal retrieval specifically to the streaming video domain, which requires tailored solutions beyond the scope of existing LLM-based systems.
>
> In summary, our work offers an in-depth analysis of the StreamingVQA task, addressing its challenges through innovations like sliding-window video encoding, video KV-cache offloading, and retrieval, culminating in a training-free method that seamlessly integrates with existing Video-LLMs for efficient and scalable solutions.
>
> [1] Liu, Haotian, et al. "Visual instruction tuning." In NeurIPS, 2024.
>
> [2] Zhang, Peiyuan, et al. "Long context transfer from language to vision." arXiv:2406.16852, 2024.
>
> [3] Zhang, Shengyu, et al. "Instruction tuning for large language models: A survey." arXiv:2308.10792, 2023.
>
> [4] Yang, An, et al. "Qwen2 technical report." arXiv:2407.10671, 2024.
>
> ---
>
> **Q2: KV-Cache Reduction Methods**
>
> Thank you for the insightful comment. Our proposed method is complementary to KV-Cache reduction techniques, as mentioned in Section 6, which lists recent advancements like quantization, token pruning, and compression. Specifically, KV-cache reduction methods can be integrated during video encoding, enabling retrieval of reduced KV-caches for question-answering. Implementing such integration mainly involves engineering considerations that fall outside the scope and contributions of this paper.
>
> ---
>
> **Q3: Generalizability**
>
> We kindly refer you to our *General Response*, where we present experiments with various Video-LLMs to address this concern.

---

> ### Author Response · Authors · 2024-11-25
> **Response to Reviewer AcRC (Part2)**
>
> **Q4: Computational Complexity (FLOPs and MACs)**
>
> We appreciate your interest in the computational complexity of our method.
>
> We ensure **fair comparisons** by using the identical Video-LLM backbone (kindly refer to *Response to Reviewer g3YS (Part3)*) under controlled streaming conditions (detailed in L430-448). Specifically, we measured the FLOPs and MACs of the base Video-LLM, Flash-VStream [1], and our external and internal retrieval methods. We analyzed **average TFLOPs and TMACs per QA over various question frequencies** in a 1-hour video, leveraging the `calflops` library [2].
>
>
>
> (a) TFLOPs / QA
> | #QAs | Baseline | Flash-VStream | ReKV (External) | ReKV (Internal) |
> |------|----------------|----------------------|------------------------|------------------------|
> | 100  | 22.4           | **15.5**                | 21.7                  | 18.5                  |
> | 200  | 12.7           | 14.1                | 11.4                  | **9.6**                   |
> | 360  | 8.5            | 13.8                | 6.8                   | **5.6**                   |
>
> (b) TMACs / QA
> | #QAs | Baseline | Flash-VStream | ReKV (External) | ReKV (Internal) |
> | ---- | -------- | ------------- | --------------- | --------------- |
> | 100  | 11.2     | **7.8**           | 10.8            | 9.2             |
> | 200  | 6.4      | 7.1           | 5.7             | **4.8**            |
> | 360  | 4.3      | 6.8           | 3.3             | **2.8**             |
>
>
>
> Key findings:
> - **Efficiency with Query Frequency:** ReKV’s efficiency improves significantly with increasing QA frequency. The video stream is encoded only once, and computed results are reused across QAs, leading to reduced per-query complexity as QA frequency rises.
> - **Comparison with Flash-VStream:** Flash-VStream outperforms ReKV at low QA frequencies (e.g., 100 QAs). However, ReKV’s complexity decreases more rapidly with increased QA frequency, primarily due to Flash-VStream’s high memory update overhead. ReKV is thus better suited for high-concurrency scenarios such as live streaming. Additionally, ReKV requires no additional training.
> - **Internal vs. External Retrieval:** Internal retrieval consistently outperforms external retrieval, reducing average FLOPs by 15.5% and MACs by 15.2%.
>
> These results underscore ReKV’s ability to balance computational efficiency and effectiveness, particularly in dynamic, high-query environments. This positions ReKV as a practical and scalable solution for streaming video understanding.
>
> We hope this clarification addresses your concerns. We are happy to incorporate these results into our final draft.
>
> ---
>
> [1] Zhang, Haoji, et al. "Flash-VStream: Memory-Based Real-Time Understanding for Long Video Streams." arXiv:2406.08085, 2024.
>
> [2] Ye, Xiaoju. "calflops: a FLOPs and params calculate tool for neural networks in pytorch framework." 2023.

---

> ### Comment · Reviewer_AcRC · 2024-11-26
> **Your answers solved my concerns.**
>
> Thank you for the authors' feedback. Your answers solved my concerns, and I raised the rating. I recommend clarifying the paper's contributions by reflecting on the feedback in a revised version.

---

> > ### Author Response · Authors · 2024-11-26
> >
> > We greatly appreciate the time you took to review our work and for raising your rating. We will carefully reflect on your suggestions and incorporate them into a revised version.

---

### Author Response · Authors · 2024-11-24
**General Response**

We sincerely thank the reviewers for their time and thoughtful feedback on our work. We are grateful for their recognition of the novelty (`t8DB`), clarity of writing (`AcRC`), and the demonstrated effectiveness (`g3YS`, `3gu5`) of our method. Below, we address the common concerns raised.

---

**Common Concern: ReKV + Different Video-LLMs**

Thank you for your interest in assessing the generalizability of our approach. To this end, we conducted experiments with additional Video-LLMs, including `Video-LLaVA-7B` [1], `LongVA-7B` [2], and `LLaVA-OV-72B` [3].

| Model                       | #frames       | MLVU dev | QaEgo4D test | EgoSchema |
|-----------------------------|---------------|----------|--------------|-----------|
| Video-LLAVA-7B             | 8             | 46.6     | 36.8         | 41.3      |
| &ensp;**+ReKV**      | 0.5 FPS&rarr;8  | **49.1 (+2.5)**     | **40.4 (+3.6)**        | **42.6 (+1.3)**     |
| LongVA-7B                  | 32            | 57.0     | 42.2         | 42.4      |
| &ensp;**+ReKV**           | 0.5 FPS&rarr;32 | **59.1 (+2.1)**    | **45.4 (+3.2)**        | **43.5 (+1.1)**     |
| LLAVA-OV-72B               | 32            | 69.7     | 53.6         | 59.6      |
| &ensp;**+ReKV**        | 0.1 FPS&rarr;32 | **73.7 (+4.0)**    | **58.4 (+4.8)**        | **62.3 (+2.7)**     |

- ReKV consistently improved performance across all models, demonstrating its robustness and adaptability.
- For LLaVA-OV-72B, the need for model sharding significantly slowed inference. To address this, we set the FPS to 0.1 to maintain efficiency during evaluation.

In the final draft, we plan to include results on additional benchmarks such as ActivityNet-QA, RVS-Ego, and RVS Movie, which leverage ChatGPT for quantitative evaluation.

[1] Lin, Bin, et al. "Video-llava: Learning united visual representation by alignment before projection." In EMNLP, 2024.

[2] Zhang, Peiyuan, et al. "Long context transfer from language to vision." arXiv:2406.16852, 2024.

[3] Li, Bo, et al. "Llava-onevision: Easy visual task transfer." arXiv:2408.03326, 2024.

---

### Meta-Review · Area_Chair_iuFo · 2024-12-17

**Metareview:**

This submission explores a training-free approach for Visual Question Answering (VQA) in Video Large Language Models (Video LLMs). It introduces a novel technique based on sliding window attention and the KV-caches which can be later used to improve the efficiency in VQA tasks. All reviewers are in favor of accepting the submission.

**Additional Comments On Reviewer Discussion:**

Initially, the reviewers raised concerns regarding the novelty of the work, the organization of the paper, as well as questions about the method and evaluation. These concerns were effectively addressed in the rebuttal discussion, leading three reviewers improving their scores.

---

### Decision · Program_Chairs · 2025-01-22

Accept (Poster)